# Safe-SD: Safe and Traceable Stable Diffusion with Text Prompt Trigger for Invisible Generative Watermarking

Submission Id: 3862

## ABSTRACT

Recently, stable diffusion (SD) models have typically flourished in the field of image synthesis and personalized editing, with a range of photorealistic and unprecedented images being successfully generated. As a result, widespread interests have been ignited to develop and use various SD-based tools for visual content creations. However, the exposures of AI-created contents on public platforms could raise both legal and ethical risks. In this regard, the traditional methods of adding watermarks to the already generated images (i.e. *post-processing*) may face a dilemma (e.g., being *erased* or *modified*) in terms of copyright protection and content monitoring, since the powerful image inversion and text-to-image editing techniques have been widely explored in SD-based methods. In this work, we propose a **Safe** and high-traceable **S**table **D**iffusion framework (namely **Safe-SD**) to adaptively implant the graphical watermarks (e.g., *QR code*) into the imperceptible structure-related pixels during generative diffusion process for supporting text-driven invisible watermarking and detection. Different previous high-cost *injection-then-detection* training framework, we design a simple and unified architecture, which makes it possible to simultaneously train watermark injection and detection in a single network, greatly improving the efficiency and convenience of use. Moreover, to further support text-driven generative watermarking and deeply explore its robustness and high-traceability, we elaborately design a $\lambda$-sampling and $\lambda$-encryption algorithm to fine-tune a latent diffuser wrapped by a VAE for balancing high-fidelity image synthesis and high-traceable watermark detection. We present our quantitative and qualitative results on two representative datasets LSUN, COCO and FFHQ, demonstrating state-of-the-art performance of Safe-SD and showing it significantly outperforms the previous approaches.

## CCS CONCEPTS

• **Security and privacy** → **Digital rights management**; • **Computing methodologies** → **Artificial intelligence**.

## KEYWORDS

Invisible Watermarking, Generative Copyright, Stable Diffusion

**ACM Reference Format:**

Anonymous Authors and Submission Id: 3862. 2024. Safe-SD: Safe and Traceable Stable Diffusion with Text Prompt Trigger for Invisible Generative Watermarking. In *Proceedings of the 32nd ACM International Conference on Multimedia (MM'24), October 28-November 1, 2024, Melbourne, Australia.* ACM, New York, NY, USA, 10 pages. https://doi.org/10.1145/nnnnnnn.nnnnnnn

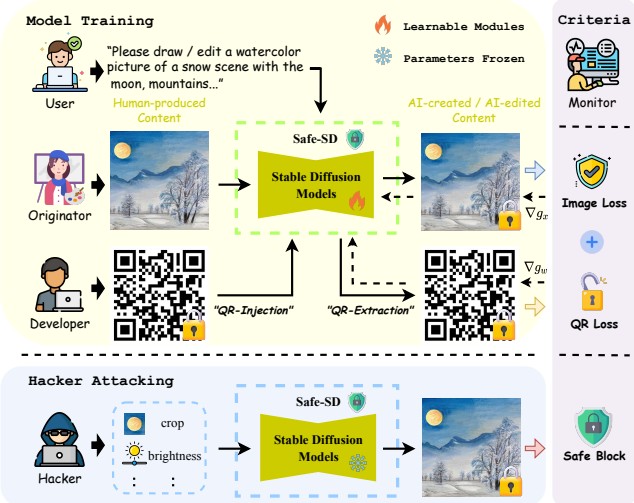

**Figure 1: The overview of our proposed Safe-SD framework. In which, different humans indicate the different roles being simulated in the AIGC environment such as *user, originator, developer, hacker* and *monitor*.**

## 1 INTRODUCTION

*"In art, what we want is the certainty that one spark of original genius shall not be extinguished."*

*– Mary Cassatt*

Recent years has witnessed the remarkable success of diffusion models [21, 44, 51], due to its impressive generative capabilities. After surpassing GAN on image synthesis [11], diffusion models have shown a promising algorithm with dense theoretical founding, and emerged as the new state-of-the-art among the deep generative models [18, 22, 29, 38, 43, 47, 50, 52, 53, 57, 64]. Notably, Stable Diffusion [48], as one of the most popular and sought-after generative models, has sparked the interests of many researchers, and a series of SD-based works have been proposed and exploited to produce plenty of AI-created or AI-edited images, such as ControlNet [67], SDEdit [40], DreamBooth [49], Imagic [27], InstructPix2Pix [3] and Null-text Inversion [41], which raises profound concerns about ethical and legal risks for AI-generated content (AIGC) being unscrupulously exposed on public platforms and raises new challenges for copyright protection and content monitoring.

These concerns may be elaborated into the following three aspects: (1) ***Originator Concern.*** An artistic work or photograph produced by the original author may be edited or modified at will by AI today and published to the public platform for commercial profit, which infringes on the interests of the originator. Take Figure 1 as an example, when a wonderful hand-crafted watercolor painting is published online by the originator, another user could download it without any restrictions and then request the SD-based model to edit the artwork through an accompanying prompt *"please edit a watercolor picture of..."*, whereas ultimately attributes the AI-created production and its ancillary value to the user and the given prompt, which may have violated the rights of the originator. If this is an

commercial advertisement or model shooting, product designs or industrial drawings, etc., it may cause more serious infringement of interests. (2) **Developer Concern.** Which means the potential risks that SD-based tools open sourced by developers may be abused by people with bad motives to engage in underground activities, such as fake news fabrication, political rumors publishing or pornographic propaganda, etc., simply by editing human characteristics (*e.g.,* replacing faces). (3) **Monitor Concern.** Which means it's extremely difficult for the monitor of online platforms to distinguish which visual contents are produced by AI and judge whether it should be safely blocked to ensure their compliance with legal and ethical standards, since the fidelity and texture of AI-created images have approached human levels. For example, a generated picture recently have won an art competition [17], which suggests humans will soon be unable to discern the subtle differences between AI-generated content and human-created content. Overall, the above concerns illustrate the fact that the emergence of powerful AI-generative tools and the lack of traceability of their generated productions may open the door to new threats such as artwork plagiarism, copyright infringement, political rumors publishing, and portrait rights infringement and so on.

To cope with the above concerns, we propose a **Safe** and high-traceable **S**table **D**iffusion framework with a text prompt trigger for unified generative watermarking and detection, *Safe-SD* for short. Note that since Stable Diffusion [48] is an open source model with most ecologically complete as well as widely used foundation models and has been applied to numerous generative tasks, we only focus on the SD-based models for invisible watermark injection and extraction, which can be further easily extended to other diffusion models such as DALL-E2 [47], Imagen [50] and Parti [64] by only replacing the weights and bias of the U-Net's parameters in diffusion models and adding a lightweight inject-convolution layer from our Safe-SD. Different from existing methods that *post-processing* [8], *injection-then-detection* [65] or are based solely on *decoder fine-tuning* [15], our proposed models have the following new features:

- Designing a unified watermarking and tracing framework, which makes it possible to simultaneously train watermark injection and detection in a single network to balance high-fidelity image synthesis and high-traceable watermark detection, greatly improving the training efficiency and convenience of use.
- Enabling to implant the graphical watermarks (e.g., *QR code*) into the imperceptible structure-related pixels, which ties the pixels of watermark to each diffusion step for high-robustness, unlike *post-processing* methods, may be easily erased or modified by image inversion or editing models.
- Supporting text-driven image watermarking and multi-watermarking scenarios, which can be applied to a wider range of downstream tasks such as: *text-to-image* synthesis, *text-based* image editing, *multi-watermarks* injection, etc.

Experiments on three representative datasets LSUN-Churches [63], COCO [36], FFHQ [25] demonstrate the effectiveness of Safe-SD, showing that it achieves the *state-of-the-art* generative results against previous invisible watermarking methods. Further qualitative evaluations exhibit the pixel-wise differences between the original images and watermarked images, and the robustness study

quantitatively evaluates the anti-attack ability, which further verifies the superiority of Safe-SD in balancing high-resolution image synthesis and high-traceable watermark detection.

## 2 RELATED WORK

**Diffusion Models.** Recent years has witnessed the remarkable success of diffusion-based generative models, due to their excellent performance in the diversity and impressive generative capabilities. These previous efforts mainly focus in sampling procedure [37, 53], conditional guidance [11, 43], likelihood maximization [28, 29] and generalization ability [18, 26] and have enabled state-of-the-art image synthesis. Stable Diffusion [48] is one of the most widely used diffusion models, due to its open source and user-friendly features, it has recently gained great attention and become one of the leading researches in image generation and manipulation.

**Image Watermarking Techniques.** To trace copyright and make AI-generated content detectable, numerous watermarking techniques have been proposed for deep neutral networks [1, 31, 32, 34, 35, 39, 42], which can basically be classified into two categories: discriminative models and generative models. In discriminative models, watermarking techniques are mainly dominated by white-box or black-box models. The white-box models [4, 7, 13, 33, 55, 56, 59, 61] need access to the models and their parameters (white-box access) in order to extract the watermarks, while the black-box models [5, 10, 20, 23, 54, 62, 66, 68] only adopt predefined inputs as triggers to query the models (black-box access) without caring about their internal details. In generative models, the previous methods mainly investigate GANs by watermarking all generative images [9, 14, 45, 70] such as binary strings embedding [14, 65, 70], textual message encoding [9] and graphic watermark injection [45]. Very recently, some researchers [15, 24, 69] have extended binary strings embedding technique into diffusion-based architecture for digital copyright protection, one of the most representative digital watermark injection methods is Stable Signature [15]. However, binary digital watermarking suffers from erasuring and overwriting threats when meeting with DDIM inversion [51], overwriting attacks [60] and backdoor attacks [6, 19].

Different from them, we explore a more secure and efficient diffusion-based generative framework *Safe-SD*, with imperceptible watermark injection module and textual prompt trigger, which is designed in a unified watermarking and tracing framework, making it possible to simultaneously achieve watermark injection and detection in a single network, greatly improving the training efficiency and convenience of use for multimedia and AIGC community. For security, the Safe-SD enables SD-based generative network to implant the graphical watermarks (e.g., QR code) into the imperceptible structure-related pixels and retain high-fidelity image synthesis and high-traceable watermark detection capabilities, which is hard to be erased or modified as the graphical watermark is tightly bound to the progressive diffusion process. For robustness, we introduce a fine-tuned latent diffuser with an elaborately designed $\lambda$-encryption algorithm for high-traceable watermarking training. Moreover, we also conduct a hacker attacking study (Sec. 4.5), by setting up 5 attack tests to evaluate the robustness of proposed Safe-SD against attacks. Note our Safe-SD methods can be easily extended to other diffusion-based models such as DALL-E2 [47],

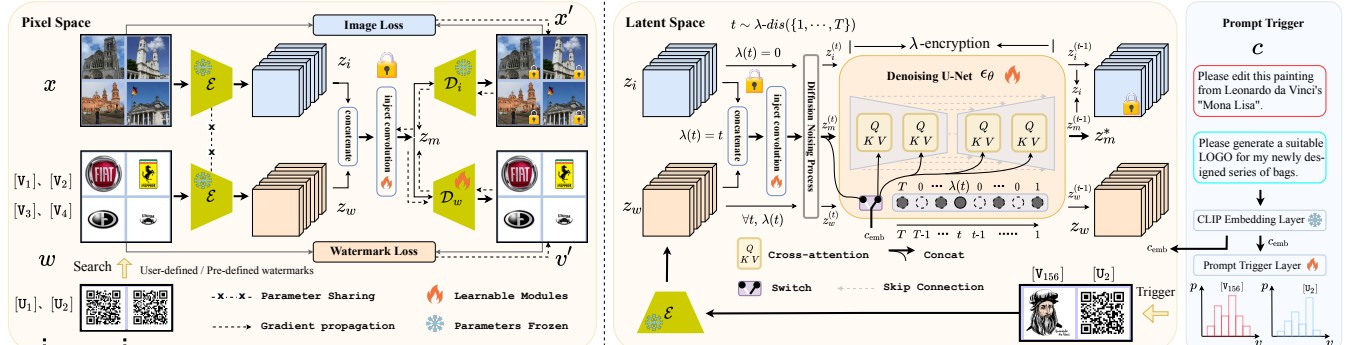

(a) **Stage1:** Pre-train graphic watermark injector/extractor    (b) **Stage2:** Fine-tune latent $\lambda$-encryption diffuser with prompt triggers

**Figure 2: The framework of Safe-SD model.**

Imagen [50] and Parti [64] by only replacing the weights and bias of the U-Net's parameters in diffusion models and adding a lightweight *inject-convolution* layer have pretrained in our Safe-SD.

## 3 METHOD

As depicted in Figure 2, Safe-SD mainly contains two stages: 1) Pre-training stage for unified watermark injector/extractor (Sec.3.1) and 2) Fine-tuning stage for latent diffuser with text prompt trigger (Sec.3.2). The former aims to train a modified SD's *first-stage-model* (with a brand new dual variational autoencoder) to obtain a unified graphic watermark injection and extraction network, whereas the latter serves as a latent diffuser with an elaborately designed temporal $\lambda$-encryption algorithm for more secure and high-traceable watermark injection. Moreover, we introduce a novel prompt triggering mechanism to support text-driven image watermarking and copyright detection scenes.

During inference, the pipeline of our proposed model is: 1) Safe-SD first accepts a text condition $c$ and an image $x$ ({"image synthesis": $x = \varnothing$; "image editing": $x$}) as inputs, and then prompt trigger $p(\cdot)$ determines which watermark $w$ should be injected based on the given condition $c$. Meanwhile Safe-SD randomly allocates a key $m \in \{0, 1\}^T$ ($T$ is diffusion steps) into the next step; 2) The encoder $\mathcal{E}$ of the *first-stage-model* first encodes the image $x$ and watermark $w$ into latent variables $z_i$ and $z_w$ respectively and then feeds them immediately into the second stage; 3) The latent diffuser first accepts the latent variables $z_i$ and $z_w$, condition $c$ and the key $m$, then performs temporal $\lambda$-encryption algorithm (**Algorithm** 1) for high-traceable watermark injection or performs condition-guided invert denoising (**Algorithm** 2) for high-fidelity image synthesis; 4) The decoder $\mathcal{D}_i$ of the *first-stage-model* then serves as a watermarker to generate the above watermarked images with $\lambda$-encryption for safe readout, and another decoder $\mathcal{D}_w$ serves as a detector to decode the injected watermark hidden from the images for detection, authentication and copyright trace.

### 3.1 Pre-training watermark injector/extractor

Our *first-stage-model* is designed to jointly train a watermark extractor $\mathcal{D}_w$ and an image generator $\mathcal{D}_i$ with invisible watermarking when they are equally fed the latent variables $z_m$ of an image mixed with watermark features. Since it is fully pre-trained to balance the two goals of simultaneously generating high-quality

images and clear watermarks, this *first-stage-model* can adapt to accept any latent mixture $z_m^*$ with $\lambda$-encryption watermarking in the second stage, to ultimately complete the dual decodings. Details of the *first-stage-model* are introduced below.

**Shared graphic encoder.** Given an input image $x$ and a randomly searched watermark $w$, $x, w \in \mathbb{R}^{H \times W \times 3}$. The shared graphic encoder $\mathcal{E}$ first projects the image $x$ and watermark $w$ into latent variables $z_i$ and $z_w$, i.e., $z_i = \mathcal{E}(x)$, $z_w = \mathcal{E}(w)$, $z_i, z_w \in \mathbb{R}^{h \times w \times d}$, where $h$ and $w$ respectively denote scaled height and width (default scaled factor $f = H/h = W/w = 8$), and $d$ is the dimensionality of the projected latent variables.

**Injection convolution layer.** Safe-SD first concatenates the projected image $z_i$ and watermark $z_w$ in the channel dimension, and then obtains the mixture features $z_m \in \mathbb{R}^{h \times w \times d}$ through a simple injection convolution layer $f_c(\cdot) : \mathbb{R}^{h \times w \times 2d} \to \mathbb{R}^{h \times w \times d}$. Formally,

$$z_m = f_c(z_i, z_w) \tag{1}$$

**Dual goal decoders.** To synchronously train a image generator $\mathcal{D}_i$ with invisible watermarking and a watermark extractor $\mathcal{D}_w$, we introduce a dual decoding mechanism with two decoder-copies from SD's *first-stage-model* (i.e., *vae* [12]), and one copy with frozen parameters $\theta_f$ and the other copy with trainable parameters $\theta_t$. Note that since decoder $\mathcal{D}_i$ plays the role of an image generator with an invisible watermark injection and has been fed to the mixture variable $z_m$, it needs to be assigned to the frozen parameter $\theta_f$ for watermarking image generation, while decoder $\mathcal{D}_w$ only serves as a watermark extractor (also with the mixed variables $z_m$ as input), therefore need to be assigned trainable parameters $\theta_t$ for watermark extraction. Formally,

$$\hat{x} = \mathcal{D}_i(z_m; \theta_f), \ \hat{w} = \mathcal{D}_w(z_m; \theta_t) \tag{2}$$

To maximize the accuracy of watermark extraction and enabling to generate high-resolution images, we set up a weighting-based loss $\mathcal{L}_{s^1}$ to supervise the entire *first-stage-model*, which can be formally represented as,

$$\mathcal{L}_{s^1} = ||x - \hat{x}||^2 + \gamma \cdot ||w - \hat{w}||^2 + \mathcal{L}_{adv} \tag{3}$$

where $\gamma$ is the weighting hyperparameter (default $\gamma$ equals 1), and $\mathcal{L}_{adv}$ denotes the adversarial training loss, which maintains the same setting as in VQGAN [12].

---

**Algorithm 1** $\lambda$-sampling based forward diffusion

**Input**: Latent image $z_i$ and watermark $z_w$, diffusion steps $T$

**Output**: $\lambda$-watermarking noise $z_m^T$, key $m$

1: $t \sim \lambda$-$dis(t) = \{\underbrace{1, 3, ..., T}_{\lambda}, \underbrace{0, ..., 0}_{T-\lambda}\}$;

2: $t \rightarrow \lambda(t)$;

3: **for** $t = 1, 2, ..., T$ **do**

4:     **if** $\lambda(t) = 0$ **then**

5:         $z_i^{(t)} = \alpha_t \cdot z_i^{(t-1)} + \sqrt{1 - \alpha_t^2}\epsilon, \ \epsilon \sim \mathcal{N}(0, I)$;

6:         $z_i^{(t)} \rightarrow z_m^{(t)}$

7:         $m_t = 0$;

8:     **else if** $\lambda(t) = t$ **then**

9:         $f_c(z_i^{(t-1)}, z_w^{(t-1)}) \rightarrow z_m^{(t-1)}$;

10:         $z_m^{(t)} = \alpha_t \cdot z_m^{(t-1)} + \sqrt{1 - \alpha_t^2}\epsilon, \ \epsilon \sim \mathcal{N}(0, I)$;

11:         $m_t = 1$;

12:     **end if**

13:     $z_w^{(t)} = \alpha_t \cdot z_w^{(t-1)} + \sqrt{1 - \alpha_t^2}\epsilon, \ \epsilon \sim \mathcal{N}(0, I)$;

14: **end for**

15: $\text{Iter}^+(z_m^{(t)}) \rightarrow z_m^T$;

16: $\text{Compose}(m_t) \rightarrow m$;

17: **return** $\{z_m^T, m\}$

---

**Algorithm 2** $\lambda$-encryption based inversion denoising

**Input**: Latent image $z_i$ and watermark $z_w$, denoising key $m$

**Output**: $\lambda$-encrypted mixture $z_m^0$, latent image $z_i^0$ and watermark $z_w^0$

1: **for** t = $T, T-1, ..., 1$ **do**

2:     **if** $m_t = 0$ **then**

3:         $z_i^{(t-1)} = \sqrt{\alpha_{t-1}}(\frac{z_i^{(t)} - \sqrt{1-\alpha_t}\epsilon_\theta^{(t)}(z_i^{(t)}, c, t)}{\sqrt{\alpha_t}}) + \sqrt{1 - \alpha_{t-1} - \sigma_t^2} \cdot$
        $\epsilon_\theta(z_i^{(t)}) + \sigma_t\epsilon, \ \epsilon \sim \mathcal{N}(0, I)$;

4:     **else if** $m_t = 1$ **then**

5:         $z_m^{(t-1)} = \sqrt{\alpha_{t-1}}(\frac{z_m^{(t)} - \sqrt{1-\alpha_t}\epsilon_\theta^{(t)}(z_m^{(t)}, c, t)}{\sqrt{\alpha_t}}) + \sqrt{1 - \alpha_{t-1} - \sigma_t^2} \cdot$
        $\epsilon_\theta^{(t)}(z_m^{(t)}) + \sigma_t\epsilon, \ \epsilon \sim \mathcal{N}(0, I)$;

6:     **end if**

7:     $z_w^{(t-1)} = \sqrt{\alpha_{t-1}}(\frac{z_w^{(t)} - \sqrt{1-\alpha_t}\epsilon_\theta^{(t)}(z_w^{(t)}, c, t)}{\sqrt{\alpha_t}}) + \sqrt{1 - \alpha_{t-1} - \sigma_t^2} \cdot$
    $\epsilon_\theta^{(t)}(z_w^{(t)}) + \sigma_t\epsilon, \ \epsilon \sim \mathcal{N}(0, I)$;

8: **end for**

9: $\text{Iter}^-(z_i^{(t)}, z_w^{(t)}) \rightarrow (z_i^0, z_w^{0,i})$;

10: $\text{Iter}^-(z_m^{(t)}, z_w^{(t)}) \rightarrow (z_m^0, z_w^{0,m})$;

11: $z_w^{0,m} \rightarrow z_w^0$ **if** $m_0 = 1$ **else** $z_w^{0,i} \rightarrow z_w^0$;

12: **return** $\{z_m^0, z_i^0, z_w^0\}$.

---

## 3.2 Fine-tuning latent $\lambda$-encryption diffuser

The *second-stage-model* mainly serves as a temporal $\lambda$-*encryption* diffuser with prompt triggering mechanism, which mainly relies on a temporal injection algorithm by accepting a binary key $m \in \{0, 1\}^T$ as *instruction-code* to control whether each diffusion step requires performing watermark injection, for cryptographic image synthesis with minor structural changes. Details of the *second-stage-model* are as follows.

**Prompt trigger.** The prompt trigger is designed to achieve non-sensitive watermark triggering, which accepts a textual *editing-* or *synthesis*-related instruction as input, by following a CLIP embedding layer and a linear prompt trigger layer, to ultimately obtain a watermark (*predefined* or *user-defined* watermark) with the highest probability for subsequent invisible watermark injection. Moreover, for stable copyright protection, Safe-SD can also support watermark injection based on special instructions, such as when given the instruction: "*Please help me edit this personal photo with my avatar watermark* [U]" and the accompanying avatar "[U]" as a personalized watermark, Safe-SD can be triggered directly with this specified watermarking LOGO. Note that in our experiments, we adopt a public LOGO dataset [1] to represent pre-defined or user-defined watermarks for the training of the Safe-SD.

**Forward diffusion with $\lambda$-sampling.** To enable the watermark to be adaptively injected into the image synthesis process with temporal diffusion and to maintain traceability, we propose the forward diffusion with $\lambda$-sampling. We first introduce the definitions of $\lambda$-*sampling* and $\lambda$-*distribution* below, and then explain how it can be used for watermark injection based on temporal encryption.

First, for a given sequence $(x_1, ..., x_N)$, the $\lambda$-*sampling* operation is defined as: randomly selecting $\lambda$ elements from the sequence with $N$ elements for sampling, and at the same time, the unsampled elements are set to 0. Thereafter the obtained discrete distribution is

---

[1] https://github.com/msn199959/Logo-2k-plus-Dataset

---

referred to as the "$\lambda$-*distribution*" corresponding to this $\lambda$-*sampling*, abbreviated as $\lambda$-$dis(\cdot)$, where,

$$\lambda\text{-}dis(i) = \begin{cases} x_i & \text{if } x_i \text{ is sampled,} \\ 0 & \text{otherwise.} \end{cases} \quad (4)$$

Then, we introduce this $\lambda$-*sampling* based temporal encryption mechanism, which aims to bind a given watermark $w$ to a diffusion synthesis process $q(z_m^{(t)} | z_i^{(t-1)}, z_w^{(t-1)})$ and simultaneously generate a binary key $m$ for traceability, as illustrated in Algorithm 1. As shown in Figure 3, when $\lambda(t)$ equals $t$, the Safe-SD is activated to perform the watermark injection process through a temporal injection cell (right side of Figure 3), which is consistent with the *first-stage-model* to ensure good generalization for watermark injection and can be formally described as,

$$z_m^{(t-1)} = f_c(z_i^{(t-1)}, z_w^{(t-1)}) \quad (5)$$

$$z_m^{(t)} = \alpha_t \cdot z_m^{(t-1)} + \sqrt{1 - \alpha_t^2}\epsilon, \ \epsilon \sim \mathcal{N}(0, I) \quad (6)$$

where $f_c(\cdot)$ denotes an aforementioned learnable injection convolution layer proposed by us for mapping the concatenation of the latent image $z_i^{(t-1)}$ and watermark $z_w^{(t-1)}$ into a latent watermarking mixture $z_m^{(t-1)}$. Whereas, when $\lambda(t)$ equals 0, the Safe-SD performs this forward diffusion simply by adding random noise $\epsilon \sim \mathcal{N}(0, I)$ to the latent vector $z_i^{(t-1)}$ of the image from the previous step, formally,

$$z_m^{(t)} = \alpha_t \cdot z_i^{(t-1)} + \sqrt{1 - \alpha_t^2}\epsilon, \ \epsilon \sim \mathcal{N}(0, I) \quad (7)$$

Note Safe-SD uses a binary value of 0 or 1 to record this forward diffusion process with $\lambda$-*sampling* and then to compose them into a binary key $m \in \{0, 1\}^T$, which will serve as readout to control the subsequent inverted denoising.

**Inverted denoising based $\lambda$-encryption.** To fine-tune the latent diffuser from *second-stage-model* to enable the input image,

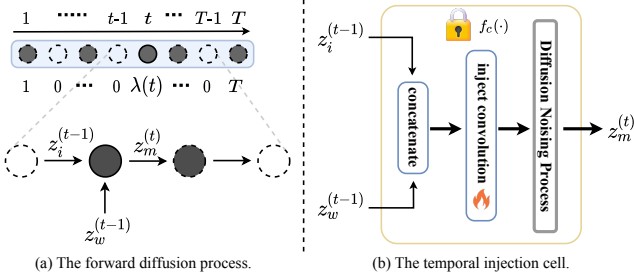

(a) The forward diffusion process.     (b) The temporal injection cell.

**Figure 3: The forward diffusion with $\lambda$-*sampling* watermarking.**

watermark and their latent mixture to be correctly denoised by an U-Net network, and to ultimately ensure high-fidelity image synthesis and watermark extraction, we propose this inverted denoising module based $\lambda$-*encryption*. Consistent with the forward process mentioned above, this inverted denoising module is controlled by an *if-else*-branched Markov chain, which is recorded by the binary key $m$ (*e.g.,* 10101101) generated above. Similarly, we first introduce the $\lambda$-*encryption* mechanism below, and then explain how it can be used for inverted denoising.

First, for a given sequence $(x_1, ..., x_N)$ and a key $m \in \{0, 1\}^N$, this $\lambda$-*encryption* is defined as: at any position where $m_i = 1$, the original data $x_i$ is modified into $x_i^*$ by superimposing a perturbation $x_\Delta$ onto $x_i$ (*i.e.,* $x_i^* = x_i \oplus x_\Delta$), while keeping the original data unchanged at other positions where $m_i = 0$, to finally obtain the encrypted sequence. The advantage of this $\lambda$-*encryption* method is that it maintains the distribution of the original data as much as possible while achieving controllable encryption.

Then, we introduce a $\lambda$-*encryption* based inverted denoising strategy, which treats the latent watermark $z_w$ as a perturbation when $m = 1$, and $z_w$ is subsequently superimposed on the latent variable $z_i$ of the image (*i.e.,* $z_m^* = z_i \oplus z_w$) by an injection convolution layer $f_c(\cdot)$ to ultimately obtain a watermarked image (*i.e.,* encrypted vector) in latent space, as shown in Figure 3(b). Formally,

$$z_m^{(t)} = z_i^{(t-1)} \oplus z_w^{(t-1)} = f_c(z_i^{(t-1)}, z_w^{(t-1)}) \qquad (8)$$

Furthermore, as illustrated in Algorithm 2, when $m = 0$, the latent variable $z_i$ of the original image is directly sent to U-Net for denoising without adding any disturbance. As shown in Figure 4, when denoising the perturbed image $z_m^{(t)}$, the watermark $z_w^{(t)}$ is simultaneously fed into U-Net for balancing image generation and watermark extraction. Note that this does not require using U-Net twice but simply by first concatenating them and then feeding them together into a shared U-Net network $\epsilon_\theta(\cdot)$ for denoising as,

$$(z_m^{(t-1)}, z_w^{(t-1)}) = \mathcal{S}_{ddim}\left(\epsilon_\theta^{(t)}(z_m^{(t)}, z_w^{(t)}|c, t)\right) \qquad (9)$$

where $\mathcal{S}_{ddim}(\cdot)$ denotes the DDIM [51] sampling strategy executed during inference, which is sampled from the predicted $\epsilon_\theta^{(t)}$ to obtain the final $z_m^{(t-1)}$ and $z_w^{(t-1)}$ (through a tensor split operation torch.chunk()). Similarly, when an unperturbed image $z_i^{(t)}$ as input, the watermark $z_w^{(t)}$ is also sent to U-Net for denoising as,

$$(z_i^{(t-1)}, z_w^{(t-1)}) = \mathcal{S}_{ddim}\left(\epsilon_\theta^{(t)}(z_i^{(t)}, z_w^{(t)}|c, t)\right) \qquad (10)$$

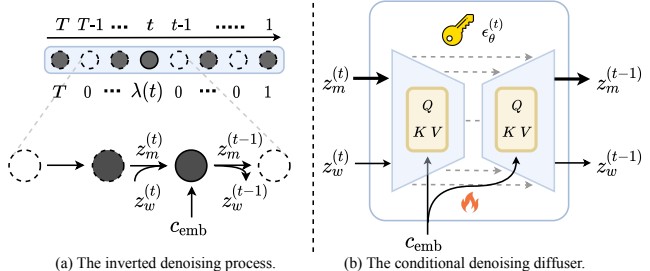

(a) The inverted denoising process.     (b) The conditional denoising diffuser.

**Figure 4: The inverted denoising based $\lambda$-*encryption* prediction.**

**Fine-tuning objectives.** To fine-tune this latent diffuser with $\lambda$-*sampling* and $\lambda$-*encryption* to adapt to the dual goal decoders from the *first-stage-model*, we set up a stepwise denoising loss,

$$\mathcal{L}_{s^2} = \underbrace{|| \epsilon - \epsilon_\theta^{(t)}(z_m^{(t)}, z_w^{(t)}) ||_2^2}_{m_t=1} + \underbrace{|| \epsilon - \epsilon_\theta^{(t)}(z_i^{(t)}, z_w^{(t)}) ||_2^2}_{m_t=0} \qquad (11)$$

where $\epsilon \sim \mathcal{N}(0, I)$ denotes standard Gaussian noise, which is consistent with Stable Diffusion [48]. Moreover, the classifier-free guidance technique [22] is also used in the training of Safe-SD.

## 4 EXPERIMENTS

### 4.1 Experimental Setting

**Datasets.** We follow [12] to pre-train the *first-stage-model* of our Safe-SD on LSUN-Churches [63], COCO [36], FFHQ [2] [25] and Logo-2K [58] datasets with image resolution $256 \times 256$, and further follow Dreambooth [49] to fine-tune the latent diffuser of the *second-stage-model* for $\lambda$−encrypted watermark injection. For the training of the text-conditional diffusion models, we follow [49] to leverage a textual prompt (*e.g., "a photo of a church with watermark* [V] *(or* [U])*")* as the guidance condition and adopt the graphical LOGOs from Logo-2K as pre-defined watermarks to finetune our Safe-SD model in our experiments. Specifically, $126, 227$ images on training set of LSUN-Churches, $63, 000$ images on training set of FFHQ and $167, 140$ watermarks on Logo-2K are utilized to train the models. During testing, $1, 000$ images and $1, 000$ watermarks are randomly composed to perform the quantitatively experimental evaluations.

**Implementation details.** We follow SD [48] to resize all the images to a resolution of $256 \times 256$, and the batch size is set to 4. The scaling factor $f$ is set to 8 and the guidance factor of the classifier-free is set to 7.5. During inference, the pre-trained CLIP embedding layer [46] is leveraged to match the suitable watermarks for adaptive prompt triggering strategy and DDIM [51] sampling is executed for final image synthesis. All the experiments are performed for 20 epochs on 2 NVIDIA RTX3090 GPUs with PyTorch framework and the optimization and schedule setups are consistent with [48].

### 4.2 Image generation quality for watermarking

**Qualitative Evaluation.** To evaluate the image generation quality and the fidelity with watermarking, we first conduct the qualitative experiments by visualizing the pixel-level differences ($\times 10$) between original image and watermarked image (marked as *W/. Watermark*), which are presented in Figure 5. Specifically, in Figure 5,

---

[2] https://github.com/NVlabs/ffhq-dataset

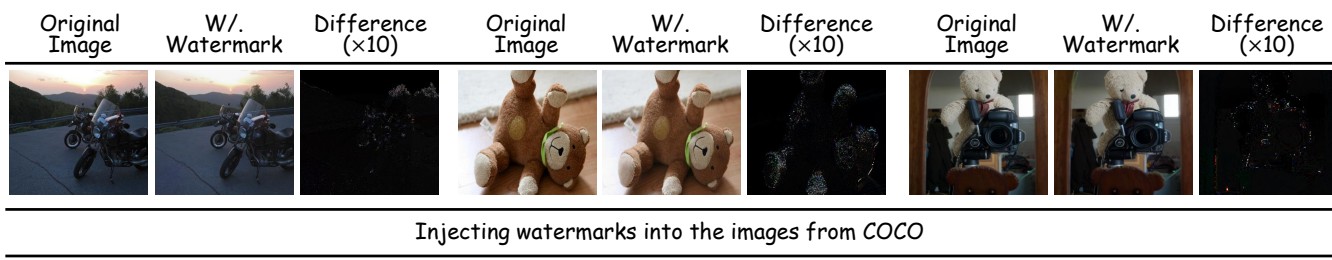

Injecting watermarks into the images from COCO

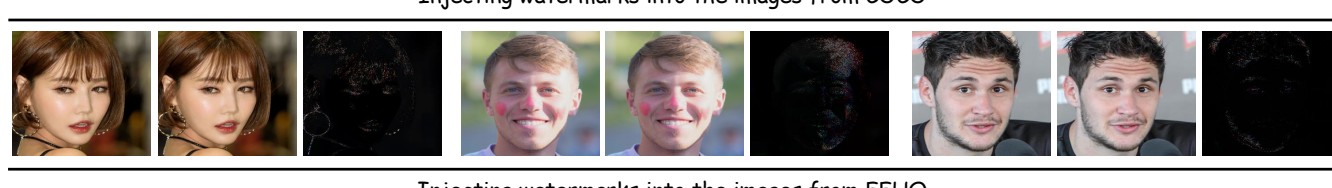

Injecting watermarks into the images from FFHQ

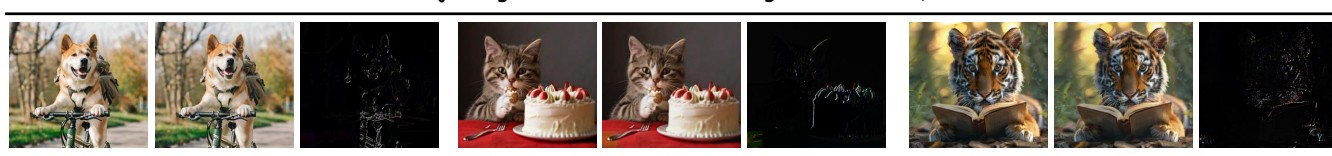

A dog is riding bicycle          A cat is eating cake          A tiger is reading book

**Figure 5: Evaluation the image quality by visualizing the pixel-level differences (×10) between original image and watermarked image (marked as W/. Watermark). Top: natural images from COCO [36]. Mid: facial images from FFHQ [25]. Bottom: text-generated images.**

| Methods | Type | PSNR ↑ | FID ↓ | LPIPS ↓ | CLIP ↑ |
|---|---|---|---|---|---|
| SSL Watermark [2022] | string | 31.60 | 19.63 | 0.261 | 84.03 |
| Baluja et.al. [2019] | graphics | 30.41 | 20.39 | 0.317 | 83.63 |
| FNNS [2021] | string | 32.71 | 19.03 | 0.243 | 85.99 |
| HiDDeN [2018] | string | 32.99 | 19.49 | 0.244 | 85.43 |
| Stable Signature [2023] | string | 31.09 | 19.47 | 0.263 | 87.90 |
| Safe-SD (Ours) | graphics | **33.17** | **18.89** | **0.232** | **88.15** |

**Table 1: The comparison results on LSUN-Churches dataset.**

| Methods | Type | PSNR ↑ | FID ↓ | LPIPS ↓ | CLIP ↑ |
|---|---|---|---|---|---|
| HiDDeN [2018] | string | 32.19 | 19.58 | 0.217 | 93.32 |
| Baluja et.al. [2019] | graphics | 29.17 | 20.85 | 0.403 | 91.93 |
| FNNS [2021] | string | 31.96 | 19.56 | 0.220 | 92.00 |
| SSL Watermark [2022] | string | 30.47 | 19.91 | 0.262 | 91.51 |
| Stable Signature [2023] | string | 30.88 | 20.33 | 0.231 | 93.01 |
| Safe-SD (Ours) | graphics | **32.73** | **19.36** | **0.215** | **93.99** |

**Table 2: The comparison results on FFHQ dataset [25].**

we respectively test Safe-SD on natural images from COCO [36] (**Top**), facial images from FFHQ [25] (**Mid**), and text-generated images (**Bottom**). From Figure 5, we can observe that: **1) All watermarked images by our Safe-SD maintain high-fidelity.** Particularly, even for challenging facial images, the watermarked results still can finely preserve the details of hair. Moreover, combined with the results of Figure 6 (additionally presenting the detected watermarks), we can notice that our Safe-SD can simultaneously balance the quality of the detected watermarks and the watermarked images. It is worth noting that compared to previous digital watermarking methods [15, 69], our Safe-SD has higher fault tolerance. For example, when some several pixels are incorrectly predicted, it will not lead to incorrect detection and authentication in our method, but in the digital watermarking method, the incorrect prediction of every binary bit (*e.g.*, "0101"→"0111") may seriously affect the final identification result. **2) There are still subtle textured differences in enlarged pixel-level, but that's almost imperceptible and well ensures traceability.** According to the enlarged (×10) pixel-wise results, it can be observed that the generative differences mainly come from visual contents with dense texture, such as hair and eyes in facial images, but note that it is almost impossible to

discern by the human eyes. That also reveals that the information hidden in the image cannot disappear, but can only be moved to an imperceptible location to ensure traceability. **3) Safe-SD is suitable for a wide variety of images and well supports text-driven generative watermarking.** As shown in Figure 5, the experiments are conducted on a wide variety of images, such as the natural images from COCO [36], facial images from FFHQ [25], and text-generated images (bottom), showing all the generated images watermarked by our Safe-SD maintain high-fidelity, which demonstrates the powerful generalization ability of our Safe-SD. Besides, Figure 6 presents more qualitative comparison results with previous graphical watermarking method [2], which further verifies the superiority of our model in balancing high-resolution image synthesis and high-traceable watermark detection.

**Quantitative Evaluation.** Following [15], we further quantitatively evaluate our approach in PSNR, FID, LPIPS and CLIP-Score metrics on LSUN-Churches and FFHQ datasets, which is shown in Table 1 and Table 2. From the results in the two tables, we can observe that our model Safe-SD achieves the *state-of-the-art* performance on all four metrics and obtains the best generative results, even with more challenging graphical watermarking, i.e.,

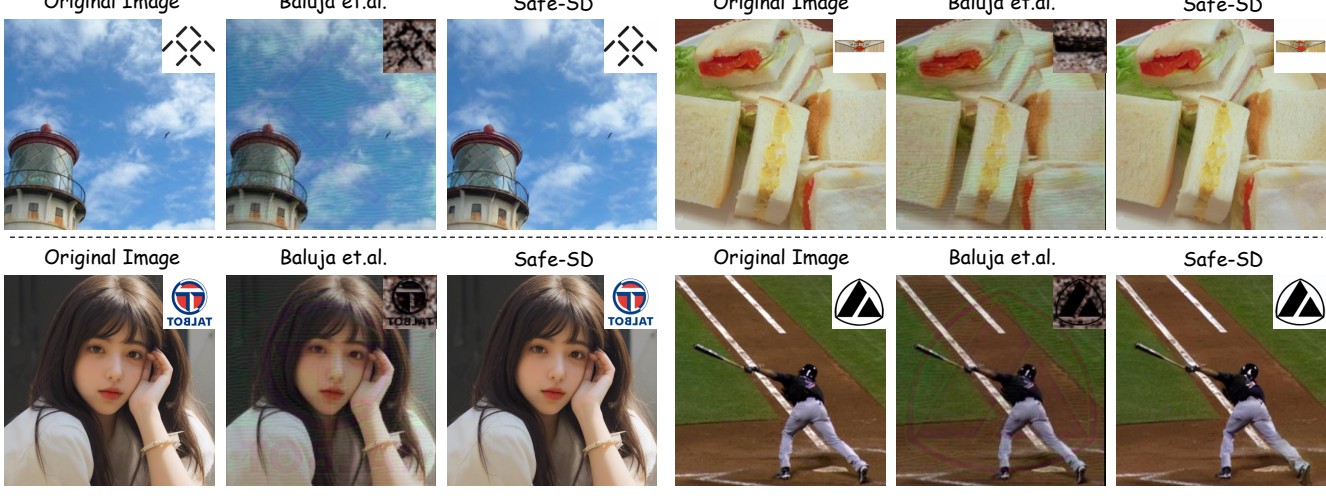

**Figure 6: Qualitative comparison results. Note the first column is the "*original image*" and "*original watermark*" (upper right corner), the second column is the "*watermarked image*" using Baluja et.al. method [2] and the "*detected watermark*" (upper right corner). The third column is consistent with the second column but with our Safe-SD approach.**

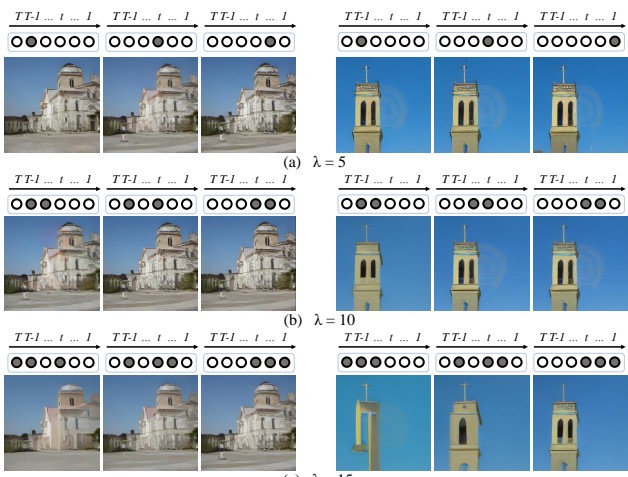

**Figure 7: The effect of the $\lambda$. Two groups of instances are presented to explore the influence of the frequency and time period of $\lambda$-encryption. Note the solid ball denotes the current $\lambda$-$dis(t)$ is not $0$.**

directly interfering with pixels, compared to string-based methods [15, 16, 30, 70]. In particular, our model outperforms Stable Signature, a recent generative work, by 6.69%, 2.98%, 11.79% and 0.28% in four metrics on the LSUN-Churches dataset, and exceeds by 5.99%, 4.77%, 6.93% and 1.05% on the FFHQ dataset, which further verifies the superiority and effectiveness of Safe-SD.

### 4.3 Explore on $\lambda$-encryption watermarking

**The frequency of $\lambda$-encryption.** To deeply explore the performance of $\lambda$-encryption in image watermarking in our approach, we perform a study on the impact of watermarking frequency $\lambda$ on image synthesis quality, as shown in Figure 7. It can be observed that with the increase of $\lambda$ (*i.e.*, from 5 to 15), the performance of generated images may be affected due to the interference of watermark

information, so we need to balance the frequency of watermark injection and the image's fidelity and finally choose $\lambda = 10$ (50 steps in total) as the appropriate watermarking frequency.

**The time period of $\lambda$-encryption.** To further explore the influence of different injection time of watermark on image synthesis quality, we also perform a study on the watermarking time period $t$, as shown in Figure 7. From Figure 7, we can observe that the earlier the injection occurs, the less high-frequency information in the image is retained in the final generative results. Particularly, when $\lambda = 15$ and the watermarking time period is in the early stage (refer to the first column of each case in Figure 7(c)), it will cause image distortion, which indicates that the watermarking unit (Figure 3) should be activated set as often as possible during the middle to end time period of latent diffusion, for better balancing watermark injection and generative effects.

### 4.4 Analysis on hyper-parameter $\gamma$

To further trading off the high-fidelity image synthesis and high-traceable watermark injection, we perform this study on hyper-parameter $\gamma$ (refer to Formula 3), as illustrated in Figure 8. From Figure 8(a), we can observe that: **1)** When the loss of image reconstruction and the loss of watermark decoding have the same weight (*i.e.*, $\gamma = 1$), both of them can steadily decrease until the model converges; **2)** When reducing $\gamma$ to make the model focus on image synthesis (*i.e.*, $\gamma = 0.1$), the loss curves of both have a significant decline in the early stage, but after that the watermark decoding becomes difficult to converge. Correspondingly, the decoded LOGO has become obviously blurred at this time; **3)** When $\gamma$ further decreases (*i.e.*, $\gamma = 0.01$), similar conclusion is further verified. Based on the above discussion, we finally choose $\gamma = 1$ to balance image synthesis and watermark decoding.

### 4.5 The robustness of watermarking

**Anti-attack test.** We conduct the anti-attack test to evaluate the robustness of our graphical watermarking against a variety

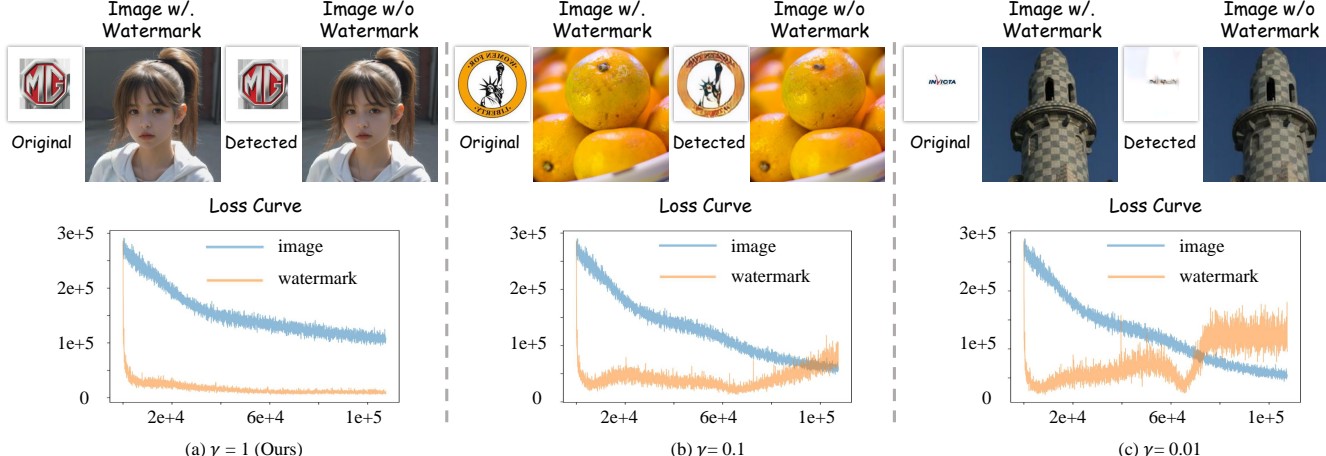

**Figure 8: The effect of the hyper-parameter γ. The generated images, watermarks and the curve of loss value are shown to qualitatively and quantitatively assess the effect of γ.**

|  | PSNR ↑ | FID ↓ | LPIPS ↓ | CLIP-Score ↑ |
|---|---|---|---|---|
| **None (Ours)** | **33.17** | **18.89** | **0.232** | **88.15** |
| Rotate 90 | 32.96 | 18.94 | 0.228 | 87.72 |
| Resize 0.7 | 32.18 | 19.11 | 0.242 | 87.03 |
| Brightness 2.0 | 30.53 | 19.77 | 0.257 | 86.09 |
| Crop 10% | 31.01 | 19.30 | 0.250 | 85.01 |
| Combined | 29.24 | 20.18 | 0.275 | 83.49 |

**Table 3: Robustness studies on LSUN-Churches dataset.**

of attacks, as shown in Table 3. Specifically, we follow Stable Signature [15] to set up 5 attack tests: **1)** *Rotate 90*, **2)** *Resize 0.7*, **3)** *Brightness 2.0*, **4)** *Crop 10%*, **5)** *Combined*. From Table 3, we can observe that our approach is robust to the various attacks. For examples, the PSNR metric under the most challenging combined attack is still higher than 29%, and the LPIPS metric is still lower than 0.28%, which demonstrate the excellent robustness of our Safe-SD. Moreover, the CLIP-Scores under all attacks are still higher than 83%, which demonstrate most of the semantic information is still retained in the watermarked images. Moreover, it can be observed that the brightness has relatively maximal impact on generation quality (*e.g.,* PSNR, FID, LPIPS), and even if the image is cropped to 10% of the original image, it still retains a high watermark recognition rate, which verifies the effectiveness of Safe-SD.

**Multi-watermarking test.** Figure 9 shows the test results of multiple watermarking. From Figure 9, we can notice that when multiple watermarks are injected at the same time, our Safe-SD still could maintain the high-quality image characteristics. Meanwhile, the two injected watermarks in Figure 9 can still be clearly extracted, demonstrating the superiority of our model in multi-watermarking scenarios.

## 5 CONCLUSION

In this paper, we have presented Safe-SD, a safe and high-traceable Stable Diffusion framework with text prompt trigger for unified

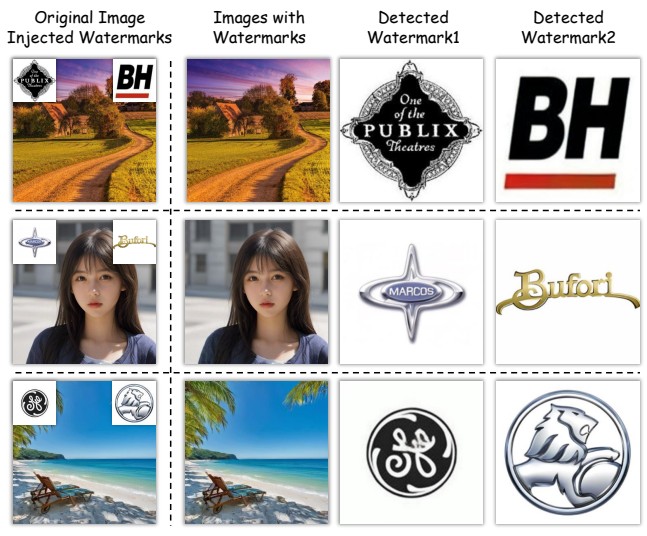

**Figure 9: Multiple watermarking evaluations.**

generative watermarking and detection. Specifically, we design a simple and unified architecture, which makes it possible to simultaneously train watermark injection and detection in a single network, greatly improving the efficiency and convenience of use. Moreover, to further support text-driven generative watermarking, we elaborately design a λ-sampling and λ-encryption algorithm to fine-tune a latent diffuser wrapped by a VAE for balancing high-fidelity image synthesis and high-traceable watermark detection. Besides, we introduce a novel prompt triggering mechanism to enable adaptive watermark injection for facilitating copyright protection. Note the proposed approach can be easily extended to other diffusion models and can adapt to various downstream tasks. Experiments on the representative LSUN-Churches, COCO, and FFHQ datasets demonstrate the effectiveness and superior performance of our Safe-SD model in both quantitative and qualitative evaluations.

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
