# OpenReview forum: "Safe-SD: Safe and Traceable Stable Diffusion with Text Prompt Trigger for Invisible Generative Watermarking"
_acmmm.org/ACMMM/2024/Conference — MM2024 Poster_

### Official Review · Reviewer_DWed · 2024-05-24

**Rating:** 5
**Confidence:** 3

**Summary:**

This paper proposes Safe-SD, a Stable Diffusion framework that is safe and highly traceable.
Safe-SD uses text prompts to enable unified generative watermarking and detection.
Safe-SD also allows simultaneous training of watermark injection and detection within a single network.

**Strengths:**

1. The paper is well organized and well structured. The figures, equations, and algorithms are very clear and easy to follow.
2. The paper develops novel a λ-sampling and λ-encryption algorithm, which can achieve high-fidelity image synthesis and high-traceable watermark detection.
3.  The paper introduces a novel prompt-triggering mechanism that enables adaptive watermark injection.

**Limitations:**

1. For lambda encryption, it treats the latent watermark as a perturbation. It keeps the original data unchanged when m=0. and adds perturbation when m=1. What if the message m is sparse and most of the bits in m are 0? Will this sparse message make an extremely subtle perturbation on the final image so that the extractor can not effectively decode the watermark image?
2. Following the above question, I find the difference in watermarked images is subtle and sparse, which is good for achieving high-fidelity watermarking quality. However, can this very subtle difference be detected by the decoder? Is there any intuition behind the scenes?

**Suitability:**

3

---

### Official Review · Reviewer_yB9C · 2024-05-24

**Rating:** 5
**Confidence:** 3

**Summary:**

This paper introduces a unified architecture for watermark injection and detection, including a λ sampling and λ encryption algorithm for text-driven generative watermarking, balancing high-fidelity image synthesis and watermark detection. A cue-triggering mechanism for adaptive watermark injection is also proposed for copyright protection.

**Strengths:**

The experimentations utilised a wide range of typical databases for testing and comparing against multiple past approaches.

The paper examines findings from both qualitative and quantitative perspectives to make the experimental clear.

**Limitations:**

The article is dense with experiments and details, yet it remains compact enough to provide a high-level introduction to the subsequent content in each section/subsection.

Although the contribution section mentioned that the proposed method supported text-driven image watermarking, the related work did not include many methods for comparison. Is it the first method that support text-driven function?

The article is well-structured when examining the individual sections, but it lacks a cohesive storyline, and the research gaps addressed are not clearly defined.

**Suitability:**

2

---

### Official Review · Reviewer_QEgF · 2024-05-25

**Rating:** 4
**Confidence:** 4

**Summary:**

This paper proposes a novel Safe-SD to hide a watermark image (QR code) during image generation. Specifically, it designs the $\lambda$-sampling and $\lambda$-encryption to fine-tune a latent diffuser for balancing high-fidelity image synthesis and watermark extraction. Experiments demonstrate its effectiveness in watermarking diffusion models.

**Strengths:**

1、The proposed Safe-SD is novel and interesting.  Protecting the copyright and safety of the diffusion model is a promising topic. Hiding images into stable diffusion via finetuning and modifying the sampling mechanism has not been fully explored by existing methods.

2、Extensive experiments demonstrate the advantages of Safe-SD compared to other image-hiding steganography methods.

**Limitations:**

1、The presentation of this paper is not sufficiently clear. Some implementation information is missing, such as how the latent code of the watermark $\boldsymbol{z}_w$ is initialized during invertible denoising based on $\lambda$-encryption.

2、The author has made many changes to Diffusion's sampling and network structure, which makes this work less scalable. I'm a little worried about whether there is enough significance in modifying it so much just to hide a picture. In fact, there are many methods of hiding images in diffusion through fine-tuning without changing the diffusion network structure [1, 2]. The authors need to compare with them and clarify the advantages of Safe-SD.

[1] A Recipe for Watermarking Diffusion Models.

[2] VillanDiffusion: A Unified Backdoor Attack Framework for Diffusion Models. NeurIPS 2023.

[3] Text-to-Image Diffusion Models can be Easily Backdoored through Multimodal Data Poisoning

3、Some important image-hiding methods have not been compared, such as [4, 5, 6]. I understand that these methods have strong fidelity, so I suggest the author compare Safe-SD with these methods in terms of security and robustness. The author should do some steganalysis to show its advantage.

[4] Large-capacity image steganography based on invertible neural networks, CVPR 2021.

[5] HiNet: deep image hiding by invertible network, ICCV 2021

[6] DeepMIH: Deep invertible network for multiple image hiding, TPAMI 2022.

**Suitability:**

2

---

### Official Review · Reviewer_6jnW · 2024-05-25

**Rating:** 3
**Confidence:** 3

**Summary:**

the authors have proposed a Safe and high-traceable Stable Diffusion framework (named Safe-SD) with a text prompt trigger for unified generative watermarking and detection. The motivation of the paper is strong. It includes emergence of powerful AI-generative tools and the lack of traceability that could lead to new threats in artwork plagiarism, copyright infringement and so on, which have ethical and legal risks for AI-generated content.

**Strengths:**

The authors presented Safe-SD, a safe and high-traceable Stable Diffusion framework with text prompt trigger for unified generative watermarking and detection. A novel prompt triggering mechanism is proposed to enable adaptive watermark injection for facilitating copyright protection.

**Limitations:**

A major effort is required to understand the proposed methodology. The complete section needs to be rewritten. One possible approach is to extensively use figures and explain each component of the methodology directly within the context of the corresponding figure. This can help clarify the process and make the section more engaging and easier to understand.

With ambiguity in methodology, it becomes tough to understand the technical functioning and contributions of the paper.

**Suitability:**

2

---

### Meta-Review · Area_Chair_nbUW · 2024-07-06

**Recommendation:** Accept (Poster)
**Confidence:** 4

**Metareview:**

This paper introduces a safe and highly traceable diffusion framework, designed to adaptively embed graphical watermarks (such as QR codes) into imperceptible structure-related pixels during the generative diffusion process. This supports text-driven invisible watermarking and detection. Unlike previous high-cost injection-then-detection training frameworks, this work presents a streamlined and unified architecture, enabling simultaneous training of both watermark injection and detection within a single network.

Reviewers feedbacks are positive: WA/WA/WA/WA. The idea is sufficiently novel. The rebuttal is effective to address concerns in experiments.